# HBV prevalence in Sub-continental countries: A systematic review and meta-analysis

**Sam Hogan** [1] *, **Andrew Page** [1], **Sameer Dixit** [2], **Kate A. McBride** [1]

**1** Translational Health Research Institute, Western Sydney University, Syndey, NSW, Australia, **2** Center for Molecular Dynamics Nepal, Kathmandu, Nepal

☯ These authors contributed equally to this work.

* 30045035@westernsydney.edu.au

## Abstract

### Background

Hepatitis B virus (HBV) is a major source of disease burden worldwide, with an estimated 296 million individuals living with infections worldwide. Although vaccine programs exist to control infections, certain sub-populations around the world continue to have very high prevalence of HBV infection.

### Methods

A systematic search of studies of HBV published after 2010 was conducted for India, Pakistan, Bangladesh, Nepal, Sri Lanka and Bhutan. Each paper was independently screened for risk of bias and inclusion. Data were extracted from included studies before being analysed to estimate pooled prevalence, and to conduct sub-group analyses. Random-effects models were used for estimating summary prevalence due to a high level of heterogeneity between studies, and funnel plots were combined with Egger's test to assess publication bias. Meta-regression was conducted to investigate sources of between-study heterogeneity.

### Results

The pooled prevalence of HBV across all studies was 3% (95% CI 0.02, 0.05). For countries with multiple studies, the pooled prevalence in India was 3% (95% CI 0.02, 0.04), in Pakistan 6% (95% CI 0.03, 0.09), in Bangladesh 5% (95% CI of 0.02, 0.12), and in Nepal 1% (95% CI 0.00, 0.08). There was some evidence of publication bias, and a high level of heterogeneity across studies. Risk of bias analysis found most studies to be of fair or moderate quality.

### Conclusions

The prevalence of HBV among countries in the sub-continent was higher than the global average, but was not as high as some other regions. Countries with greater numbers of displaced persons had higher prevalence of HBV, with a wide range of prevalence between sub-populations likely reflecting differential uptake, and implementation, of vaccination programs.

**Data Availability Statement:** "All underlying data in this review is freely available based on the previous publications included in this review. The dataset created for use in this systematic review and meta-

analysis is available in the Supporting Information files."

**Funding:** The authors received no specific funding for this work.

**Competing interests:** The authors have declared that no competing interests exist.

## Introduction

Hepatitis B is an infection caused by the hepatitis B virus (HBV) and can lead to severe complications in infected individuals. HBV infections are a major cause of health problems worldwide, with both chronic and acute infections presenting different symptoms and complications. In 2015, HBV caused approximately 887,000 deaths worldwide and there were an estimated 296 million people living with a chronic infection [1], although these numbers differ geographically [2]. Acute HBV infections have no specific treatment, with the majority of treatments aiming to reduce symptoms and managing the discomfort of the infected individual. Symptoms of acute HBV infections can include dehydration, diarrhoea and vomiting. Individuals can also develop acute liver failure which can lead to death [3]. Around 5% of adults with acute HBV go on to develop a chronic infection [4], which can cause cirrhosis of the liver and in some cases, hepatocellular cancer [2]. Children, especially newborns, are extremely vulnerable to developing chronic HBV infections, as the majority of those infected within the first year of life will go on to develop a chronic infection. Although the chance of developing chronic infection reduces with age, 30–50% of those infected with HBV before the age of six will develop a chronic infection [1, 4]. This is especially problematic, as one of the most common routes of transmission of HBV infections is vertical transmission from mother-to-child or direct transmission from an infected child to others via exposure to infected blood from the child 1, 2]. Other methods of transmission include exposure to infected blood or other bodily fluids, sexual transmission and intravenous drug use via sharing of needles or use of unsterilised needles [2, 5]. Contaminated razors can also be a method of transmission, which can increase prevalence where barbershops play important roles both socially and culturally [6].

Although a vaccine is available for HBV that provides protection from infection, high levels of vaccination coverage are not ubiquitous globally. The rate of HBV infection among general populations also can vary from country to country, as vaccination regimes and protocols differ between settings. Different regions around the world have varying prevalence of HBV, with the World Health Organization (WHO) Western Pacific region and African region reporting the highest burden of infection (approximately 116 million and 81 million infected respectively) [1]. Many developing countries began implementing childhood HBV vaccination programs relatively recently, which have shown some success in reducing the prevalence of HBV in younger populations such as children and young adults [7, 8]. In the Indian Subcontinent vaccination programs have been introduced in the selected countries from the early 2000s it is important to examine changes in prevalence which may be occurring, especially as there have been different prevalence levels reported from certain subgroups which are higher than those of national averages and other international contexts [9, 10].

In this systematic review and meta-analysis, we sought to ascertain the prevalence of HBV infection among populations within subcontinental countries. For studies which included HBV vaccination status as a variable, vaccination rates were also examined as a secondary aim. Additionally, subgroups were assessed to determine which factors increase risk of HBV infection and reduce the chance of being vaccinated for HBV. Pre- and post-vaccination cohorts were also examined to assess differences in HBV prevalence that may be attributable to vaccination programs within settings, where data was available.

## Methods

### Study setting

The countries chosen for inclusion in this study were those of the Indian Subcontinent, namely India, Pakistan, Bangladesh, Nepal, Sri Lanka and Bhutan. The Maldives were also included in

the search strategy, however no relevant papers were located. The Indian Subcontinent was chosen as the overall region of focus for this review, rather than a single setting, as the countries are all densely populated low- and middle-income countries, share a close national history, and there is relatively free movement across borders for many of these countries. Thus, examining vaccination coverage within the populations of interest was a secondary aim of this review. General populations from both rural and urban populations were also chosen for inclusion in this systematic review, as these populations have different risk factors that may increase or decrease risk of HBV infection.

## Study design and protocol registration

The protocol of this systematic review and meta-analysis was designed based on the Preferred Reporting Items for Systematic Reviews and Meta-Analysis Protocols (PRISMA-P) Guidelines [11]. The protocol for this systematic review was registered in Prospero prior to the initial search (protocol registration number of CRD42020215743).

## Ethics approval

All studies included in this review had been given the appropriate ethics approval for their study setting when selected. As this study is a systematic review and meta-analysis, ethics approval for this review was not sought.

## Search strategy

A systematic search was conducted using multiple databases; PubMed, Embase, Medline and the Cochrane Library. Web of Science was also used, but all articles returned from this search were duplicated in the results from the other databases. The search strategy restricted results to only those published after 2010, to ensure recency and increase the relevance of the papers. Some of the countries of interest had previously had systematic reviews focusing on HBV which included results up to 2010, so this rationale was also considered when defining this restriction.

The search strategy used a combination of keywords, search symbols and MeSH terms to strengthen the search (S1 Checklist). The search was conducted by the first author (SH), and the results from each database combined (Fig 1). Following this, the titles were scanned to assess relevance, which was done by assessing whether the terms "HBV" or "prevalence" were included in the titles. Titles which did not include one of these two terms were excluded from the next phase, where abstracts were independently assessed by each reviewer (SH, AP and KAM) for relevance. When assessing risk of bias, full text reviews were conducted on papers included following the abstract screen, which were again independently conducted by SH, AP and KAM. The search was most recently conducted on 31$^{st}$ May 2023.

## Inclusion and exclusion criteria

Inclusion criteria were studies conducted in countries in the sub-continent, published after 2010, among participants of any age, and reported HBV prevalence established via a valid testing method (ranging from rapid test kits to lab confirmation). Studies were restricted to English language articles, and with full text available. No specific study design was specified for inclusion, however due to the nature of the outcome of interest, most studies were cross-sectional study designs. A secondary aim of the study was also to assess vaccination coverage within the study populations, however this was not included in many of the papers which focussed on prevalence. The only exclusion criterion was that participants could have no other

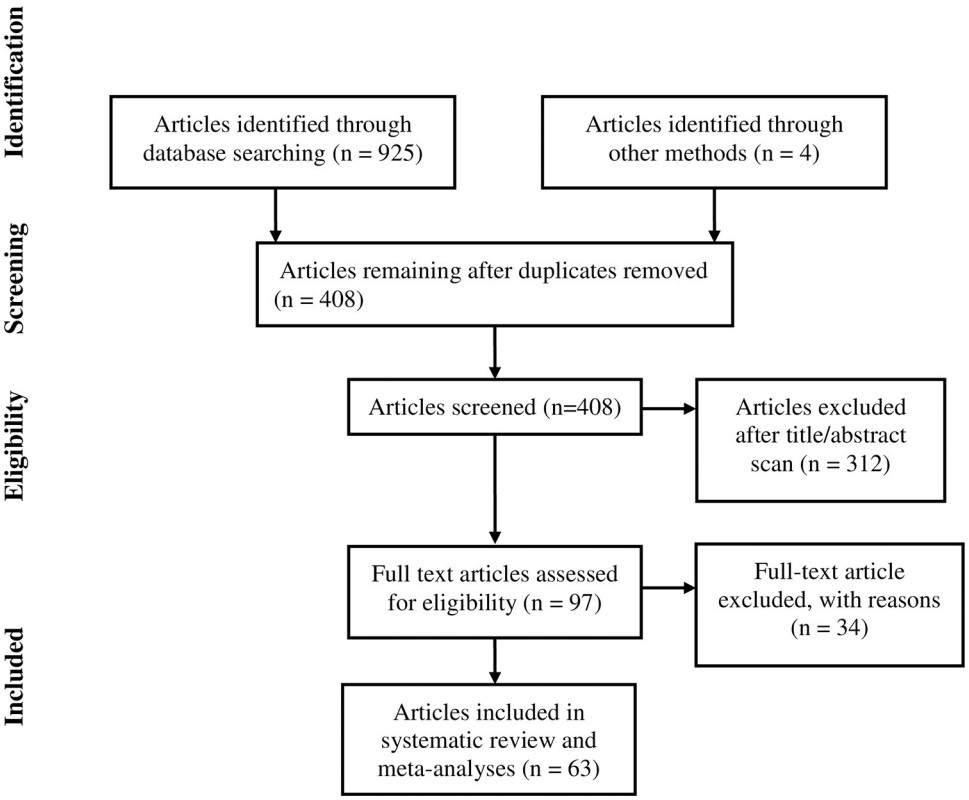

**Fig 1. PRISMA flowchart showing the selection process of the studies.**

underlying health issues (e.g., thalassemia patients, hepatocarcinoma patients) as these populations would likely not be representative of the general populations of identified countries.

## Data extraction

The data extracted from each study included study design, laboratory method of HBV diagnosis, age, sex, geographic location, literacy level, and vaccination status (where available), to allow for subgroup analysis by these variables.

## Quality assessment

Following selection of studies that met the inclusion criteria, data extraction was performed for all relevant variables. The Joanna Briggs Checklist for Prevalence Studies [12] was used to assess the quality of the studies and risk of bias (ROB). This tool was chosen as it has been shown to be a valid method of assessing prevalence studies, while other tools such as the Cochrane Risk of Bias tool are not appropriate for this type of study design. All studies were assessed for ROB by one author (SH), and two other authors (AP and KAM) assessed 50% of the papers each. Once ROB had been assessed, data analysis was performed on the selected papers. There was some disagreement between reviewers when classifying the quality of the studies (disagreement on 24/63 papers, 38%), however these issues were resolved by the 3rd independent reviewer on the papers they were not originally allocated to conduct independent ROB (KAM or AP).

## Data synthesis and analysis

For pooled prevalence, random effects models were used as it was likely that variance between identified studies was due to more than only selection bias or sampling errors. Additionally, it was thought that a random effects model was more appropriate for estimates which included a larger population. There were several features which differed between the studies, such as geographic location, study population, sample size and in some cases diagnostic tests. Heterogeneity between studies was assessed using the $I^2$ statistic, which can be used to interpret the percentage of variation between studies which is due to heterogeneity rather than pure chance [13]. High $I^2$ percentages represent a greater level of heterogeneity, however this is not always a methodological weakness, especially in systematic reviews incorporating different study designs [13]. Confidence intervals (95%) were also given for the meta-analyses, with weights of each of the studies also shown. Publication bias and small-study effects were assessed using, funnel plots and the Egger's test statistic. Subgroup analysis and meta-regression were used to assess sources of between study heterogeneity between studies. Meta-regression assessing the impact of a number of variables on the likelihood of being infected with HBV were used to identify common risk factors within the datasets. Meta-regressions were performed using country, sex, location, study quality, and year of publication, and source population of the studies. Literacy level was also used for a smaller meta-regression using those papers in which it was a recorded variable. Subgroup analysis was possible using sex, country, geographic location (i.e., rural or urban), age groups, study quality, and literacy levels. These categories were also used for meta-regressions when possible. Age group definitions varied across the included studies, therefore subgroups were created around common age-group definitions across studies. Similarly, literacy and illiteracy were clearly defined in some studies [9, 14, 15] but not in others. Biological sex was the only variable which was consistently present in each of the studies, however as some studies focussed exclusively on either males or females, not all studies were included in the analysis for this subgroup analysis.

Data analyses were conducted using R [16] and RStudio [17] using the "metafor" package [18], while some figures were visualised from the same dataset using the Stata software package. These packages were chosen as they were best able to handle the dataset, while the Stata package was able to produce clearer and more detailed figures.

## Patient and public involvement

This review was conducted on de-identified secondary data, thus contacting the participants involved would be highly impractical. However, the findings of this paper will be discussed with participants in future projects focussing on Hepatitis B to help determine which findings are most relevant to the public. This will ensure that any recommendations from these projects will be focussed towards the benefit of the people most at risk of becoming infected with HBV, and those currently living with HBV infections.

## Results

### Prevalence of HBV

Of the 63 papers that met the inclusion criteria, 26 were based in India, 22 in Pakistan, 8 in Bangladesh, 4 in Nepal, 2 from Sri Lanka and 1 from Bhutan. The pooled prevalence of HBV for all included studies was 3% (95%CI 0.02, 0.05) (Table 1, Fig 2). The prevalence of HBV for Sri Lanka and Bhutan was <1% (95%CI 0.00, 0.01) and 1% (95%CI 0.01, 0.02) respectively, which were based on a low study count for each country. The pooled prevalence of HBV was

**Table 1. Subgroup analysis assessing pooled prevalence of HBV and sources of heterogeneity.**

| | Variable Category | Included studies (n) | Pooled Prevalence (95% CI) | $I^2$ | % P-value | Cumulative Tau-Squared estimate* |
|---|---|---|---|---|---|---|
| **Sex** | Male | 19 | 4% (0.02, 0.06) | 99% | 0.00 | |
| | Female | 19 | 3% (0.02, 0.04) | 97% | <0.01 | |
| | Both | 49 | 4% (0.02, 0.05) | 100% | 0.00 | |
| | | | | | | 1.9236 |
| **Geographic Location** | Urban | 38 | 3% (0.02, 0.05) | 100% | 0.00 | |
| | Rural | 17 | 5% (0.03, 0.08) | 98% | <0.01 | |
| | | | | | | 1.8755 |
| **Study quality** | Poor | 7 | 7% (0.02, 0.23) | 100% | 0.00 | |
| | Fair | 38 | 3% (0.02, 0.05) | 99% | 0.00 | |
| | Good | 18 | 4% (0.02, 0.07) | 100% | 0.00 | |
| | | | | | | 1.7796 |
| **Publication date** | Year | 63 | 4% (0.02, 0.05) | 100% | 0.00 | |
| | | | | | | 1.7452 |
| **Country** | India | 26 | 3% (0.02, 0.04) | 100% | 0.00 | |
| | Pakistan | 22 | 6% (0.03, 0.09) | 100% | 0.00 | |
| | Bangladesh | 8 | 5% (0.02, 0.12) | 100% | 0.00 | |
| | Nepal | 4 | 1% (0.00, 0.02) | 99% | <0.01 | |
| | Sri Lanka | 2 | <1% (0.00, 0.01) | N/A | N/A | |
| | Bhutan | 1 | 1% (0.01, 0.02) | N/A | N/A | |
| | | | | | | 1.5022 |
| **Source population** | | | | | | |
| | General population | 17 | 5% (0.03, 0.08) | 100% | 0.00 | |
| | Blood donors | 9 | 1% (0.01, 0.03) | 100% | 0.00 | |
| | Other | 37 | 3% (0.02, 0.05) | 100% | 0.00 | |
| | | | | | | 1.3531 |

***Tau-squared estimated from multivariate meta-regression.** For the univariate and multivariate regressions, the intercept was set at studies which included both variables (for example, the referent level used for sex and geographic location was "Both").

3% (95%CI 0.02, 0.04) in India, 6% (95%CI 0.03, 0.09) in Pakistan, 5% (95%CI 0.02, 0.12) in Bangladesh, and 1% (95%CI 0.00, 0.08) in Nepal.

The majority of the included studies were cross-sectional studies, with the exception of one prospective study, two retrospective studies and eight studies with poorly described study design.

There was also marked within- and between-country differences in HBV prevalence. For example, the highest prevalence for India was recorded by Kuriakose and Ittyachen [19] who were investigating a rural area of Kerala. Although the sample population was relatively small, the HBV prevalence was 30% (95% CI = 0.24, 0.37), which is far higher than the pooled prevalence of 3% (95% CI = 0.02, 0.04).

The total population of all included studies was 759,524 individuals from across the 6 countries, with 18,452 HBV infections. Heterogeneity between the studies was very high, with $I^2 = 100\%$ and $\tau^2 = 2.0073$ for the pooled prevalence of all studies, and was also very high for country-specific pooled prevalence. This high level of heterogeneity is likely due to the highly varied population sizes/characteristics and locations of the studies, as many studies were conducted using a similar design.

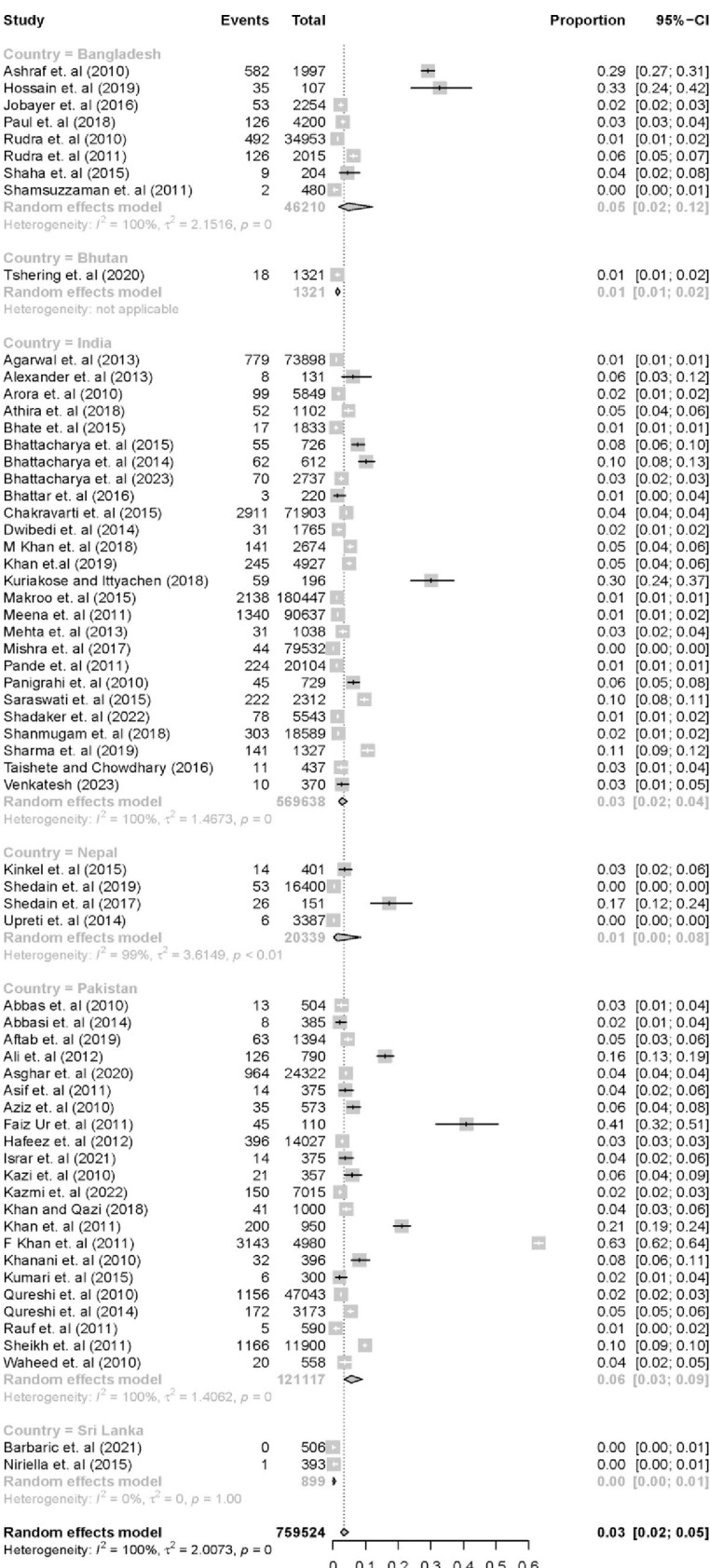

**Fig 2. Forest plot of pooled prevalence stratified by country.** The prevalence from each study is shown, as are the weights for the final overall pooled prevalence.

## Vaccination coverage

Vaccination coverage was assessed in several of the studies, however vaccination coverage was predominantly ascertained via participant self-report. The majority of the studies assessing vaccination coverage found the completion rate for the full 3 dose course was less than 90%, with the pooled average being 59% (95% CI 0.42, 0.75) across these studies [7, 8, 20–26]. There were also some studies which included data from study populations born before national vaccination programs were initiated compared to those born after the programs were introduced [8]. For example, a study based in Bhutan found that the prevalence of HBV was far lower in the population who had been born after the vaccination program had been implemented (2.0% in the pre-vaccination program group, 0.5% in the post-vaccination group [8]. Additionally, in a Nepal-based study, the prevalence of HBV was markedly lower in the post-vaccination cohort of children when compared to a pre-vaccination cohort (0.28% in pre-vaccination children, 0.13% in post-vaccination program children) [7].

## Subgroup analysis

The pooled prevalence of HBV among adult males was 4% (95%CI 0.02, 0.06), established using the random effects model. For adult females, including pregnant women, the pooled prevalence of HBV was 3% (95%CI 0.02, 0.04). The pooled prevalence of HBV among the rural only populations was 5% (95%CI 0.03, 0.08), while urban populations had a lower prevalence of 3% (95%CI 0.02, 0.05). When comparing literate vs illiterate populations, the pooled prevalence of HBV was 9% (95%CI 0.03, 0.23) compared to 18% (95%CI 0.13, 0.23), although the number of articles which recorded this variable was small (n = 3). The results of the subgroup analyses are shown in Table 1.

Some of the subgroups examined had a markedly higher prevalence of HBV than the national average for their corresponding general population, although some subpopulations reflected the expected prevalence for the countries in which they were located. For example, in a study which examined individuals in Northern Pakistan which had been displaced by conflict [9] the prevalence of HBV was far higher (21% compared to the pooled prevalence of 6%).

$\tau^2$ was used to determine variation of the true effect within subgroups examined in the meta-analysis. Meta-regressions suggested that the main source of between study variation was country of study setting (Tau-Sq = 1.5022), followed by the source population (Tau-Sq = 1.3531), and study quality (Tau-Sq = 1.7796) (Table 1). These variables were associated with the biggest change in Tau-Sq within the multivariate model. The variables selected within the multivariate regression also accounted for the majority of between study variation.

## Risk of bias

The majority of the studies included in this review were of fair quality (n = 38), with only a few rated as being poor (n = 7) (Table 2). Eighteen of the papers were rated as being of good quality, as these papers reported on all key aspects of their methodology. There was some evidence of publication bias, as indicated by funnel plots for all subgroups (S1 Fig), with Hedge's G values indicating significant asymmetry.

## Discussion

The Indian subcontinent is a region with a low overall prevalence of HBV infection, however wide variation in HBV prevalence remains between, and within, countries. Understanding these regional variations is important to understand the performance of immunisation programs and potential priorities for health system strengthening within this geographic region.

**Table 2. Summary characteristics of the studies included in this review.** The variables listed are first author, year of publication, location, country of setting, population type, HBV positive (n), total population, HBV prevalence, and Risk of Bias assessment.

| Authors | Publication Year | Location | Country | Population type | HBV Positive (n) | Total Study Population (n) | Prevalence | RoB |
|---|---|---|---|---|---|---|---|---|
| Abbas et al. [20] | 2010 | Karachi | Pakistan | General Population | 13 | 504 | 2.60% | Fair |
| Abbasi et al. [27] | 2014 | Sukkur | Pakistan | Male barbers | 8 | 385 | 2.10% | Fair |
| Aftab et al. [14] | 2019 | Punjab | Pakistan | Pregnant women | 63 | 1394 | 4.52% | Fair |
| Agarwal et al. [28] | 2013 | New Delhi | India | Blood donors | 779 | 73898 | 1.05% | Fair |
| Alexander et al. [21] | 2013 | Tamil Nadu | India | General Population | 8 | 131 | 6.10% | Fair |
| Ali et al. [15] | 2012 | Waziristan | Pakistan | General Population | 126 | 790 | 15.94% | Fair |
| Asghar et al. [29] | 2020 | Sindh | Pakistan | General Population | 964 | 24322 | 3.96% | Good |
| Arora et al. [30] | 2010 | Haryana | India | Blood donors | 99 | 5849 | 1.70% | Fair |
| Ashraf et al. [31] | 2010 | Dhaka | Bangladesh | General Population | 582 | 1997 | 29.00% | Good |
| Asif et al. [22] | 2011 | Mirpukhas | Pakistan | Medical Students | 14 | 375 | 3.70% | Poor |
| Athira et al. [32] | 2018 | Puducherry | India | Blood donors | 52 | 1102 | 4.71% | Fair |
| Aziz et al. [33] | 2010 | Sindh | Pakistan | General Population | 35 | 573 | 6.10% | Good |
| Barbaric et al. [34] | 2021 | Colombo and Jaffna | Sri Lanka | Transgender women | 0 | 506 | 0.00% | Good |
| Bhate et al. [35] | 2015 | Maharashtra | India | General Population | 17 | 1833 | 0.90% | Good |
| Bhattacharya et al. [36] | 2015 | Andaman and Nicobar Islands | India | General Population of these islands | 55 | 726 | 7.50% | Fair |
| Bhattacharya et al. [23] | 2014 | Nicobar Islands | India | General Population | 62 | 612 | 10.10% | Fair |
| Bhattacharya et al. [37] | 2023 | Odisha | India | Tribal population | 70 | 2737 | 2.56% | Good |
| Bhattar et al. [38] | 2016 | New Delhi | India | Hospital patients | 3 | 220 | 1.30% | Fair |
| Chakravarti et al. [39] | 2015 | Delhi | India | Blood samples | 2911 | 71903 | 4.05% | Poor |
| Dwibedi et al. [40] | 2014 | Odisha | India | Tribal population | 31 | 1765 | 1.70% | Good |
| Faiz Ur et al. [41] | 2011 | Peshawar and Abbotabad | Pakistan | Patients with hepatitis | 45 | 110 | 40.91% | Poor |
| Hafeez et al. [42] | 2012 | Lahore | Pakistan | Paramilitary personnel | 396 | 14027 | 2.80% | Fair |
| Hossain et al. [43] | 2019 | Mymensingh | Bangladesh | Hospital patients | 35 | 107 | 32.71% | Fair |
| Israr et al. [44] | 2021 | Swabi, Khyber Pakhtunkwa | Pakistan | Pregnant women | 14 | 375 | 3.73% | Good |
| Jobayer et al. [45] | 2016 | Dhaka | Bangladesh | Male overseas workers | 53 | 2254 | 2.35% | Fair |
| Kazi et al. [46] | 2010 | Karachi | Pakistan | Prisoners | 21 | 357 | 5.90% | Good |
| Kazmi et al. [47] | 2022 | Azad Jammu and Kashmir | Pakistan | University students and employees | 150 | 7015 | 2.14% | Fair |
| Khan and Qazi [48] | 2018 | North Waziristan | Pakistan | Internally displaced persons | 41 | 1000 | 4.10% | Fair |
| Khan et al. [9] | 2011 | Malakand Division | Pakistan | Internally displaced persons | 200 | 950 | 21.05% | Fair |
| F Khan et al. [49] | 2011 | Punjab | Pakistan | HBsAg Positive blood samples | 3143 | 4980 | 62.93% | Fair |
| M Khan et al. [50] | 2018 | Ladakh | India | Villagers | 141 | 2674 | 5.27% | Good |
| S Khan et.al [51] | 2019 | Meerut | India | Hospital patients | 245 | 4927 | 4.97% | Fair |
| Khanani et al. [52] | 2010 | Karachi, Sangar and Larkana | Pakistan | Men who have sex with men | 32 | 396 | 8.31% | Fair |
| Kinkel et al. [53] | 2015 | Nepalgunj, Biratnagar and Kathmandu | Nepal | PWIDs | 14 | 401 | 3.49% | Good |
| Kumari et al. [54] | 2015 | Karachi | Pakistan | Pregnant women | 6 | 300 | 2.00% | Fair |
| Kuriakose and Ittyachen [19] | 2018 | Kerala | India | Households in rural Kerala | 59 | 196 | 30.10% | Fair |
| Makroo et al. [55] | 2015 | New Delhi | India | Blood donors | 2138 | 180447 | 1.18% | Fair |
| Meena et al. [56] | 2011 | New Delhi | India | Blood donors | 1340 | 90637 | 1.47% | Fair |

*(Continued)*

**Table 2.** (Continued)

| Authors | Publication Year | Location | Country | Population type | HBV Positive (n) | Total Study Population (n) | Prevalence | RoB |
|---|---|---|---|---|---|---|---|---|
| Mehta et al. [57] | 2013 | Rajkot | India | Pregnant women | 31 | 1038 | 2.98% | Fair |
| Mishra et al. [58] | 2017 | Gujarat | India | Blood samples | 44 | 79532 | 0.06% | Fair |
| Niriella et al. [59] | 2015 | Mahara and Welikada Prisons | Sri Lanka | Prisoners | 1 | 393 | 0.00% | Fair |
| Pande et al. [60] | 2011 | New Delhi | India | Pregnant women | 224 | 20104 | 1.11% | Fair |
| Panigrahi et al. [61] | 2010 | Behrampur, Ganjam and Orissa | India | Blood donors | 45 | 729 | 6.17% | Fair |
| Paul et al. [24] | 2018 | Country-wide | Bangladesh | Children | 126 | 4200 | 3% | Good |
| Qureshi et al. [62] | 2010 | Country-wide | Pakistan | General Population | 1156 | 47043 | 2.46% | Good |
| Qureshi et al. [25] | 2014 | Balochistan, Sindh and Punjab | Pakistan | Mothers and Children | 123 | 1561 | 7.88% | Fair |
| Rauf et al. [63] | 2011 | Swat | Pakistan | Internally displaced persons | 5 | 590 | 0.85% | Poor |
| Ray Saraswati et al. [64] | 2015 | Delhi | India | PWIDS | 222 | 2312 | 9.60% | Fair |
| Rudra et al. [65] | 2010 | Khulna | Bangladesh | Blood donors | 492 | 34953 | 1.41% | Fair |
| Rudra et al. [66] | 2011 | Mymensingh | Bangladesh | General population | 126 | 2015 | 6.25% | Fair |
| Shadaker et al. [26] | 2022 | Punjab | India | General population | 78 | 5543 | 1.40% | Fair |
| Shaha et al. [67] | 2015 | Dhaka | Bangladesh | General Population | 9 | 204 | 4.40% | Fair |
| Shamsuzzaman et al. [68] | 2011 | Gaibandha | Bangladesh | Pregnant women | 2 | 480 | 0.04% | Fair |
| Shanmugam et al. [69] | 2018 | Tamil Nadu | India | General Population | 303 | 18589 | 1.63% | Fair |
| Sharma et al. [70] | 2019 | Himachal Pradesh | India | Villagers | 141 | 1327 | 10.63% | Good |
| Shedain et al. [71] | 2019 | Kathmandu | Nepal | Pregnant women | 53 | 16400 | 0.32% | Fair |
| Shedain et al. [10] | 2017 | Dolpa | Nepal | Mothers and Children | 26 | 151 | 17.22% | Fair |
| Sheikh et al. [72] | 2011 | Balochistan | Pakistan | General population | 1166 | 11900 | 9.79% | Good |
| Taishete and Chowdhary [73] | 2016 | Maharashtra | India | Health care workers | 11 | 437 | 2.52% | Poor |
| Tshering et al. [8] | 2020 | Country-wide | Bhutan | General population | 18 | 1320 | 1.36% | Good |
| Upreti et al. [7] | 2014 | Country-wide | Nepal | Children | 6 | 3387 | 0.18% | Fair |
| Venkatesh et al. [74] | 2023 | Bhubaneshwar, Odisha | India | General population | 10 | 370 | 2.70% | Fair |
| Waheed et al. [75] | 2010 | Lahore | Pakistan | Hospital patients | 20 | 558 | 3.58% | Fair |

Nepal, for example, began widespread distribution of the HBV vaccine in 2002, although the prevalence of HBV within the country was already relatively low, with prevalence estimated to be between 2–4% [7]. The estimated prevalence of HBV in Nepal is now estimated to be 0.9% [71]. In comparison, other countries within the subcontinent have a higher prevalence, with India (4–7% prevalence) being the highest [70]. Pakistan has recorded a prevalence of 3–5% in their general population [14], while Bangladesh has an estimated prevalence of 5.4% [43]. Bhutan and Sri Lanka both have lower prevalence, with an estimated prevalence of approximately 2% in each country [8, 59]. These lower prevalence levels are likely due to various geographic and cultural features. For example, Sri Lanka shares no land borders with other countries, while Bhutan is relatively isolated due to the dense forest and mountainous terrain which comprises the majority of the country.

Various subgroups within the general population of each of these countries have also been shown to have a higher prevalence of HBV due to a variety of risk-taking behaviours, for example use of intravenous drugs [76]. Although some risk-taking behaviours are self-determined

by the individuals (such as using intravenous drugs), others may be related to lifestyle or living conditions which are outside of the control of the individual, such as being displaced by internal conflicts [9, 48].

The prevalence of HBV among the populations/sub populations included in this systematic review varied significantly. While some of the study populations reflected the expected prevalence for the country in which the study was situated, other subpopulations from the same country had far higher prevalence of HBV. Additionally, vaccination programs appear to have had variable success across these settings, with some reporting coverage far lower than the targeted proportion. However, some of the included studies did show that vaccination programs have been successfully introduced in some of the settings such as in Bhutan [8] and Nepal [7]. In these studies, there was a clear reduction in the prevalence of HBV in pre- and post-vaccination program groups. HBV prevalence in general is lower at present than it has been historically, however there are still many "at-risk" groups with far higher prevalence than the general population. Usually, these populations are disadvantaged in some way, for example being displaced by war [48].

In general, studies set in urban locations had a lower HBV prevalence than those in more rural areas. This could be due to a number of factors such as decreased access to healthcare, differences in education level and higher frequency of other "at-risk" behaviours. Study participants who were illiterate were also far more likely to have an HBV infection, which may be tied with overall education level and health literacy, as they may be unaware of common behaviours increase risk of HBV.

When examining which population groups studies focussed on, those which focussed on the general population showed a higher pooled prevalence than any other group. Studies were classified as examining the 'general population' if they did not restrict the sample or did not focus on a specific feature when sampling. Some groups were suspected of being at higher risk of HBV infection due to a variety of factors, but were still representative of the general population. However, the studies which examined specific sub populations focussed on groups which would not be representative of the wider population as a whole, such as medical students [22], healthcare workers [73] or the general population of a specific group of remote islands [23, 36]. This meant the variation between prevalence of HBV among each group varied greatly, as some reasons that these populations were not representative of the wider population placed them at greater risk of either becoming infected, or having greater protection (e.g., remoteness). The overall pooled prevalence of the 'general population' group was 5% (95% CI 0.03, 0.08) using the random effects model, while the "other" population subgroup had a pooled prevalence of 3% (95% CI 0.02, 0.05). The sample population with the lowest pooled prevalence of HBV were blood donors (1%, 95% CI 0.01, 0.03), which was expected as these individuals are often more health conscious.

Studies classified as "Fair" or "Good" after risk of bias assessment had the same prevalence levels of 3% (95% CI 0.02–0.05) with similar confidence interval ranges. However, studies classified as "Poor" had a far higher prevalence of 7% (95% CI 0.02, 0.23). This may be due to selection of study populations not representative of the general population (e.g., hospital patients with known hepatitis). However, the settings of some of these studies were likely to have impacted the quality of the studies themselves and may have made data collection more difficult, for example the studies focussing on rural areas or populations who had been displaced from their normal area of residence [9]. It is unlikely that the difference is due to measurement error as almost all the studies utilised an enzyme-linked immunosorbent assay (ELISA) test to confirm HBV infection status, with some also utilising polymerase chain reaction (PCR) tests to additionally confirm diagnoses. Sampling issues may have affected several of the studies, as many were based on convenience samples. While follow up is not normally

relevant in cross-sectional studies, sample selection should still be adequately described in all papers. The main issue with many of the papers was how representative the study population was of the wider population the studies were set in. Additionally, some papers had poor reporting on how study populations were recruited, what the response rates were and how sampling was undertaken.

It is likely there is some level of publication or sample bias, as the high level of asymmetry shown in each of the funnel plots suggest that the prevalence from many of the studies fell outside the expected range. However, while some of these studies were potentially published as they indicated highly statistically significant results, this does not necessarily mean that all of the studies were. For example, the studies which were based on "at-risk" or vulnerable populations possibly showed higher prevalence for reasons other than those identified within the studies. Additionally, some of the studies with larger sample sizes showed a lower prevalence than that of the general population. However, some of the studies with the largest population sizes specifically examined blood donations. This population is likely to be healthier than the general population, so this may have affected observed results.

Other limitations common among the included studies were difficulties in recruiting a representative sample. Many of the studies focussed on specific subgroups rather than the general population, limiting generalisability beyond those subgroups. While these subgroups are often at an increased risk of HBV, this is typically due factors not common to the general population (for example, refugee or intravenous drug user samples). However, some of the studies included in this review had substantial sample sizes, with smaller studies contributing a lower weight in pooled estimates.

## Conclusion

The results of this systematic review and meta-analysis of Hepatitis B virus infections within the countries of the Indian Subcontinent show that although the overall prevalence of HBV is decreasing in many countries, certain sub-populations remain at an increased risk of HBV infection. These groups in particular are those who have been displaced from their homes, as well as those who live in rural areas. Men also had a slightly higher risk of HBV infection than women, possibly a reflection of the different behaviours between sexes within country settings. Although there are some clear risk factors that increase the likelihood of HBV infection, the overall aetiology remains complicated, and vaccination programs remain critically important to help reduce the prevalence further.

## Supporting information

**S1 Checklist. Checklist of items to include when reporting a systematic review or meta-analysis.**
(DOC)

**S1 Table. Summary of the terms used for the systematic search of electronic databases.**
(DOCX)

**S1 Data. Contains a link to the dataset used for this review.**
(XLSX)

**S1 Fig. Funnel plot showing the chance of publication bias having an effect on study results.** The shape of this plot indicates the large amount of variation between studies included in this review, and indicates the possibility that some publication bias exists.
(DOCX)

## Acknowledgments

The authors would like to acknowledge the previous researchers and authors involved in the studies selected for this review. Additionally, they would like to acknowledge the support from their respective organisations.

## Author Contributions

**Conceptualization:** Sam Hogan, Sameer Dixit, Kate A. McBride.

**Data curation:** Sam Hogan.

**Formal analysis:** Sam Hogan, Kate A. McBride.

**Investigation:** Sam Hogan, Kate A. McBride.

**Methodology:** Sam Hogan, Andrew Page, Kate A. McBride.

**Project administration:** Sam Hogan.

**Resources:** Sam Hogan.

**Supervision:** Andrew Page, Sameer Dixit, Kate A. McBride.

**Writing – original draft:** Sam Hogan, Andrew Page, Sameer Dixit, Kate A. McBride.

**Writing – review & editing:** Sam Hogan, Andrew Page, Sameer Dixit, Kate A. McBride.

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
