## [Decision Letter · Decision Letter 0]

27 Jul 2023

PONE-D-23-15736HBV prevalence in Sub-continental countries: a systematic review and meta-analysis

PLOS ONE 

Dear Dr. Hogan,

Thank you for submitting your manuscript to PLOS ONE. After careful consideration, we feel that it has merit but does not fully meet PLOS ONE’s publication criteria as it currently stands. Therefore, we invite you to submit a revised version of the manuscript that addresses the points raised during the review process.

Thank you for submitting your manuscript to PLOS ONE. After careful consideration,the manuscript,

HBV prevalence in Sub-continental countries: a systematic review and meta-analysisPLOS ONE has now been assessed.

We invite you to revise your paper. When your revision is ready, please submit the updated manuscript and a point-to-point response if applicable.

We look forward to receiving your revised manuscript.

Kind regards,

Sana Riaz

Academic Editor

PLOS ONE

3. We notice that your supplementary figures are uploaded with the file type 'Figure'. Please amend the file type to 'Supporting Information'. Please ensure that each Supporting Information file has a legend listed in the manuscript after the references list.

Reviewers' comments:

Reviewer's Responses to Questions

**Comments to the Author**

1. Is the manuscript technically sound, and do the data support the conclusions?

Reviewer #1: Yes

Reviewer #2: Yes

Reviewer #3: Yes

2. Has the statistical analysis been performed appropriately and rigorously? 

Reviewer #1: Yes

Reviewer #2: Yes

Reviewer #3: Yes

3. Have the authors made all data underlying the findings in their manuscript fully available?

Reviewer #1: Yes

Reviewer #2: Yes

Reviewer #3: Yes

4. Is the manuscript presented in an intelligible fashion and written in standard English?

Reviewer #1: Yes

Reviewer #2: Yes

Reviewer #3: Yes

5. Review Comments to the Author

Reviewer #1: The research paper is well written and presented, overall, study provides valuable insights into the prevalence of HBV in the mentioned countries, and the suggestions provided should help improve the clarity and presentation of the findings. Make sure to double-check the statistical values and revise the manuscript thoroughly for minor formatting errors before finalizing it.

Reviewer #2: The present manuscript entitled "HBV prevalence in Sub-continental countries: a systematic review and meta-analysis" is a meta analysis of research after 2010. HBV is an important problem in subcontinent countries and governments are trying to overcome it. Present paper may be helpful for a better understanding of HBV. Although it is written good, however, languistic errors are found. I recommend to improve english language.

Reviewer #3: 1. Page number 7, Line number 159-177 (Data synthesis and analysis)

o The use of random effects models is justified, but the reason for selecting these models could be further elaborated. Also, the statement "variance between identified studies was due to more than only selection bias or sampling errors" needs clarification as it is unclear what other sources of variance are considered.

o The I2 statistic is mentioned without any interpretation of what it means. The reader should be informed about the level of heterogeneity and its implications for the meta-analysis results.

o There is a mention of the software packages used for data analysis, but it doesn't clarify why two different packages were used. It should be explained why specific packages were chosen and if there was any specific reason to use Stata for visualizing figures.

2. Page number 18, Line number 264: The sentence is incomplete (Remains…)

3. Page number 19, line number 292: please correct the sentence (“at-risk” groups…)

4. Page number 19, line number 298: please correct the sentence (“at-risk” behaviors…)

5. Page number 20, line number 303: please correct the sentence (Focused and focusing used in the same sentence)

6. Page number 20, line number 304, 305, 306, 308, 309, 310: please correct the sentence (the word “population” is repeated 2 and 3 times)

7. Page number 22, line number 337, 338: please correct the sentence, it does not have any sense.

6. PLOS authors have the option to publish the peer review history of their article (what does this mean?). If published, this will include your full peer review and any attached files.

Reviewer #1: **Yes: **Dr. Tariq Javed Department of Pharmaceutical Sciences GCU Lahore

Reviewer #2: No

Reviewer #3: No

---

## [Author Response · Author response to Decision Letter 0]

19 Nov 2023

We thank the reviewer for their comments and suggestions for improving the manuscript, and hope that the amendments made help alleviate the issues noted in the review.

---

## [Editor Report · Decision Letter 1]

28 Nov 2023

HBV prevalence in Sub-continental countries: a systematic review and meta-analysis

PONE-D-23-15736R1

Dear Dr. Sam Hogan,

We’re pleased to inform you that your manuscript has been judged scientifically suitable for publication and will be formally accepted for publication once it meets all outstanding technical requirements.

Kind regards,

Sana Riaz

Academic Editor

PLOS ONE

Additional Editor Comments (optional):

The author has revised the manuscript as per reviewers suggestions
---

## [Editor Report · Acceptance letter]

30 Nov 2023

PONE-D-23-15736R1 

HBV prevalence in Sub-continental countries: a systematic review and meta-analysis 

Dear Dr. Hogan:

I'm pleased to inform you that your manuscript has been deemed suitable for publication in PLOS ONE. Congratulations! Your manuscript is now with our production department. 

Kind regards, 

on behalf of

Dr. Sana Riaz 

Academic Editor

PLOS ONE